# "Hearing the pupils' voices through my own struggles": A qualitative study of return to work among school counselors who are breast cancer survivors

Inbar Levkovich[1]*, Lahav Rosman[1], Christina Signorelli[2,3]

1 Faculty of Graduate Studies, Oranim Academic College, Kiryat Tiv'on, Israel, 2 School of Clinical Medicine, UNSW Medicine & Health, Discipline of Paediatrics, UNSW Sydney, Kensington, NSW, Australia, 3 Kids Cancer Centre, Sydney Children's Hospital, Randwick, NSW, Australia

* inbar.levkovich@oranim.ac.il

**Data Availability Statement:** Data cannot be shared beyond individual quotations in the paper due to participants' lack of consent for the publication of their full transcripts. The interviews

## Abstract

For breast cancer survivors, returning to work is an important step for their personal, financial, and psycho-social recovery. Returning to work as a school counselor can be particularly challenging because of the demands of their job and stress at work. This qualitative study examines return to work among school counselors who are breast cancer survivors. In-depth, semi-structured interviews were conducted with 28 survivors of breast cancer stages I–III between the ages of 32 and 55, and up to ten years after the completion of chemotherapy. Interviews focused on the discovery of the illness, treatment period, ramifications of the diagnosis on various aspects of life, and implications for work. Using thematic analysis of the data collected, analysis of the findings revealed three key themes: 1) "Everyone is replaceable": The significance of disruptions in work continuity for school counselors who are breast cancer survivors. 2) "From Zero to a Hundred": Challenges Faced by Counselors in Returning to Work after Breast Cancer Recovery.3) "It's hard to listen to counselees' problems when I am immersed in my own crisis": How surviving breast cancer affects return to work among school counselors. Findings highlight the unique needs of these counselors and the challenges they face upon returning to work. The study discusses recommendations for school principals including training, advocacy, and awareness to support survivors and improve their return to work.

## Introduction

Breast cancer is one of the most prevalent forms of cancer among women in the Western world, including in Israel [1]. In recent years, breast cancer recovery and survival rates have increased [2, 3]. In developed regions, overall, 5-year survival from breast cancer is well over 80% [4]. As a result, professionals have become more aware that breast cancer survivors continue to experience physical and emotional symptoms for extended periods after their treatment ends [5]. These symptoms can include fatigue, emotional distress, sleep disturbances and

contain sensitive data, and this decision aligns with the ethics committee of Oranim College. Please contact Miss Shaulov, administrative coordinator of the research authority in Oranim College for any further information: research@oranim.ac.il.

**Funding:** The authors received no specific funding for this work.

**Competing interests:** The authors have declared that no competing interests exist.

significant harm to the survivors' quality of life and employment status [6, 7]. Breast cancer survivors exhibit distinct needs concerning social aspects, including work and daily activities [8]. Returning to one's work role is a significant component of post-cancer life, as it promotes continued social interactions, self-esteem, financial stability, and psychological well-being [9, 10]. Nonetheless, resuming work after breast cancer presents challenges related to the recurring effects of the disease or treatments, such as fatigue and pain, as well as workplace-related obstacles like insufficient support, discrimination, termination, and social stigma [11, 12].

After completing treatment and upon returning to work, most breast cancer survivors report that the workplace provides them with a sense of self-esteem, social belonging and economic security [6, 13]. Isaksson et al. [14] reported that 54% of cancer survivors between the ages of 34 and 66 returned to their place of work within 24 months after the end of their treatment. In literature review of 25 articles, the findings from all the studies examined consistently revealed a decline in both work engagement and work ability among breast cancer survivors [6]. Women who returned to work reported negative factors, including a lack of empathy on the part of their employer, a desire for early retirement, psychological problems, and fatigue due to long working hours which also limited their capacity for work, compared to healthy controls [15, 16]. They also reported positive factors. For example, work was found to play an important role in survivors' lives and to help them return to normal. Moreover, work distracted them from thoughts about illness and financial security [9, 10, 17]. Nevertheless, survivors must readjust to their workplace or else change their occupation [18].

For breast cancer survivors, returning to work is an important step in their financial and psycho-social recovery. This step constitutes proof of their return to their familiar routine, signaling their return to healthy life [9, 18–20]. Research in this area has examined return to work among breast cancer survivors working in different professions [12, 21]. A study examining breast cancer survivors who work in the health professions (e.g., doctors, nurses, allied health professionals) found that 89% returned to work within a year. Most reported that they had to make adjustments after returning to work, such as reducing their hours, limiting the number of shifts they worked, restricting physical labor and reducing travel [22].

To apply for a license in educational counseling in Israel, candidates must possess a master's degree in educational counseling and a teaching certificate, as per the guidelines of the Ministry of Education's Counseling Division [23]. Furthermore, school counselors must adhere to the code of ethics established by the Educational Counseling Board [23]. School counselors are integral to the educational system, working full-time with students, educational staff, and parents. Their responsibilities extend beyond school hours, encompassing emergency situations, delivering lectures, and attending meetings. Although there is no specific policy for counselors recovering from illnesses such as cancer, those with over two years of service typically enjoy employment tenure protection under the Ministry of Education [24]. The current study examined the perspectives of school counselors who survived breast cancer and returned to the workplace. The job of school counselor is multidimensional and includes working individually with students and parents as well as enlisting the help of the entire school staff and relies on psychosocial functioning to perform in this role [23]. Because of the demands of their job, school counselors are under a great deal of stress at work. Tension and anxiety at work are liable to cause fatigue, problems in functioning and decreased emotional availability [25]. Despite their increased risk of work stress, to the best of our knowledge, no research has examined school counselors in this context.

This study examines the following primary research question: How do school counselors who survived breast cancer perceive their return to the workplace? The study also examines a number of secondary questions: What challenges do these school counselors face upon

returning to work? What factors inhibit their reintegration into the workplace and what factors promote it?

## Methods

The study used a qualitative-phenomenological approach [26], which sought to gain an in-depth understanding of the subjective meaning of the perspectives of school counselors who have survived breast cancer upon their return to work. Guidelines for ensuring rigor in qualitative research were followed and the researchers adhered to the COREQ checklist for reporting qualitative data [27].

### Research participants

The research population included 28 school counselors in Israel. Inclusion criteria were as follows: school counselor who work in schools in Israel; diagnosed with breast cancer stages I to III; completion of cancer treatment (1 to 5 years post-chemotherapy) and the achievement of cancer remission; returned to work the in the last five years preceding the study; Hebrew speaker.

The research participants ranged in age from 32 to 55 (M = 41.5, SD = 7.34) and worked at schools across Israel. Their experience in the field of school counseling ranged from 2 to 32 years (M = 0.68, SD = 8.05). All research participants were employed in public schools overseen by the Ministry of Education. Among the 28 participants, 11 served in high schools, 7 in middle schools, and 10 in elementary schools. Prior to their illness, all participants were full-time employees. All of them underwent surgery and chemotherapy, and 78% also had radiation therapy. The time of return to work relative to the time of remission ranged from 1 to 5 years (M = 4.53, SD = 2.43).

### Research design

Participants were recruited using social media platforms, specifically Facebook, where an explanation of the study and the researchers' contact information were provided. Additionally, snowball sampling played a role in the recruitment process, with interviewees referring their colleagues to us.

Participation in the study was voluntary, and the interviewers promised to maintain participants' anonymity and confidentiality. All participants gave their informed consent before joining the study and were given assurances that their names and schools would remain confidential [28]. The Israeli education system resumed in-person learning in 2022. The interviews for this study were conducted via private video room using the video-conference platform Zoom in January through March 2023, with the intention of including a diverse range of participants, particularly those who are geographically remote. Both interviewers were female and experienced in conducting qualitative research. I.L is a psychotherapist (PhD) and L.R a school counsellor (MA). Before conducting interviews, interviewers were required to engage in a reflective process [29]. This involved introspecting their own identities, social standings, assumptions, and life experiences, including any related to their own illness or that of their family members. They also considered how these factors might influence their interactions with interviewees. This preparatory step was essential for maintaining integrity in the research process. The interviews lasted 45–60 minutes. Each interview was recorded and later transcribed verbatim. The participants were informed they were free to discontinue their participation at any time. None of the participants chose to do so.

## Research instrument

The qualitative data were collected through one-on-one, in-depth, semi-structured interviews, each conducted with an interviewee only once. The interview enabled the participants to express their thoughts and feelings and gave them the opportunity to process their experience again. The interview was conducted based on a semi-structured interview guide that was divided into major key areas. The guide was flexible and gave opportunities for asking spontaneous questions that were likely to advance the dialogue and enhance the experience of the research participants [26]. The interview guide was compiled based upon the research literature and focused on the following content areas: discovery of the illness, treatment period, ramifications of the diagnosis on various aspects of life, implications for work. Sample questions include: "How did your work colleagues and the school principal respond when they heard about your diagnosis?" "What was the most significant challenge in returning to work?" "Have there been any changes in what you need from the system?" and "Did you receive any response to these needs?". Data collection was continued until the point of 'theoretical saturation' was achieved, wherein conducting additional interviews did not yield any new material or information that could contribute to the analysis process.

## Data analysis

After the data were collected and the interviews were transcribed, the data underwent thematic content analysis. Firstly, to achieve a state of immersion and obtain a thorough understanding of the data, all gathered information was meticulously read multiple times. Subsequently, the researcher (I.L.) employed an open coding approach, meticulously scrutinizing each line of the interview transcripts while making detailed notes. This procedure aimed to identify the initial units of meaning derived from the data and facilitate the assignment of subthemes. Furthermore, to ensure rigor, a second researcher (L.R.) reviewed the broader themes that emerged from the data and engaged in discussions with I.L. Moreover, the researchers adopted axial coding techniques, gradually uncovering the contextual and content-related associations between themes and subthemes (I.L.). To consolidate the meanings and provide structure, all interviews were compared, and names were assigned to the themes. Following this, an examination of the interrelationships among the initial codes was conducted, sorting them into higher-order theoretical codes. Finally, the researchers integrated (I.L. and L.R.) the results by conceptually rearranging the core themes that emerged from the data and situating them within their respective contexts. This analytical process facilitated the analysis and synthesis of substantial amounts of data, enabling the generation of abstractions and interpretations. In this stage, categories that did not refer to the main themes were eliminated in order to focus on the main themes [26, 28].

**Trustworthiness.** The methodological trustworthiness of this study was ensured through several rigorously applied strategies. Initially, the fidelity of the interview data was maintained by transcribing the responses verbatim. This meticulous approach allowed for an accurate and reliable return to the original narrative accounts as needed. Further bolstering the study's trustworthiness, the research team employed established credibility criteria, specifically those that emphasize the presentation of data as detailed and contextualized portrayals of varied realities [30]. In alignment with these criteria, this study employed comprehensive, in-depth interviews. These interviews were designed to facilitate a free and thorough expression of the interviewees' perceptions, particularly regarding the experiences of educational counselors who have encountered cancer. The depth and comprehensiveness of these interviews were instrumental in enabling the research team to assert that they had attained a holistic and authentic comprehension of the experiences and interpretations of the participants [30].

**Ethical considerations.** The study and consent procedure were approved by the Institutional Review Board (IRB) Ethics Committee of Oranim College in accordance with the Declaration of Helsinki (Authorization No. 107). All participants provided written informed consent and were informed that study publications would not disclose any identifying information. The research team explicitly stated during the interviews that no names or identifying details would be published, reiterating this condition in the written informed consent form that participants signed. In this article, pseudonyms or the term "the participants" are used to refer to the individuals involved, ensuring their anonymity. All study data were stored safely, and only the researcher had access to the data. All personal data were coded, so that the identities of the participants remained confidential.

## Results

The findings are discussed in this section as they emerged from the interviews with the school counselors. Three main themes emerged from this study: 1) "Everyone is replaceable": The significance of disruptions in work continuity for school counselors who are breast cancer survivors. 2) "It's hard to listen to counselees' problems when I am immersed in my own crisis": How surviving breast cancer affects return to work among school counselors. 3) "What you see from there you can't see from here": Changes in perspective among school counselors. Table 1 contains a summary of findings and sub-topics within each of the three key themes.

### Theme 1: "Everyone is replaceable": The significance of disruptions in work continuity for school counselors who are breast cancer survivors

While being treated for cancer, the counselors reported feeling that they were unable to continue working at the schools as they had in the past. Most of the participants described this as a period of adjustment in which they were forced to discontinue their ordinary routine. Some of the counselors described feeling impatient, troubled and discouraged by staying at home for such a long time during their treatment. Many of them noted that they missed their fixed routine, their job at school and their sense of belonging and of being needed.

**Table 1. Summary of findings.**

| Theme | Sub-topic |
|---|---|
| Theme 1: "Everyone is replaceable": The significance of disruptions in work continuity for educational counselors who are breast cancer survivors | • Difficulty staying at home for long period of time<br>• Harmful impact on professional identity<br>• Drop in sense of self-esteem |
| Theme 2: "From Zero to a Hundred": Challenges Faced by Counselors in Returning to Work after Breast Cancer Recovery | • Ambivalence regarding return to work<br>• Lack of understanding from the educational staff<br>• Meeting employers' expectations and lack of flexibility in response to their needs<br>• Physical symptoms that make their work more difficult; drop in emotional availability |
| Theme 3: "It's hard to listen to counselees' problems when I am immersed in my own crisis": How surviving breast cancer affects return to work among educational counselors | • They felt less patient and emotionally available for the students.<br>• Sometimes it seemed to them that the problems the students shared were less compared to the difficult experiences they had with the disease.<br>• The counselors experienced a deep sense of guilt and frustration, as they grappled with their perceived inability to provide counseling as effectively as they had in the past. |

*I can certainly say that it was very difficult at first. I told my husband I wanted to go back to work. I didn't care about my health condition. You stay home with the kids. I'm not willing to talk about this anymore. I could not tolerate it (Sarah, age 36).*

Most of the school counselors understood that their job could not remain unfilled until they were able to return to work. Yet, when they were told that a new counselor had come to replace them, they could not believe that a replacement had already been found for them and they felt betrayed. They described feeling insecure, insulted, scorned and not appreciated by the system. All the study participants described that whilst they were at home they were preoccupied with worries about the fate of their careers.

*When the treatments made it impossible for me to come to work, another counselor came to the school. I couldn't believe it. I was so jealous. The illness took so many things away from me, and now this? What if she is good, even very good, and they won't want me back? (Dana, age 55).*

The counselors also reported feeling that their professional identity had been insulted. Their safe place had been undermined and they felt worthless because they were not helping anyone, as they were formerly accustomed to. Participants described that this instigated a process of self-search and self-definition, in which they examined what defined them beyond being school counselors.

*I felt detached and remote. That was terrifying, because if I'm not a counselor, what am I? I was shocked by how important this is to me. Another school year is beginning and I'm at home. What will I do now? (Rivka, age 45).*

A minority of the school counselors reported that while they managed to keep in touch, sustaining these connections over an extended period was challenging. Some counselors experienced feelings of support, concern, and warmth from others, whereas others perceived pity, fear, and a sense of distance, suggesting that the relationships felt obligatory rather than stemming from genuine concern. One counselor shared,

*They would call occasionally to check on me, inquire if I needed anything, or ask how I was managing. It was comforting to know someone was thinking of me. At times, I sensed their pity through the phone, which made it difficult. Therefore, I sometimes opted to respond with WhatsApp messages when I didn't have the energy for phone conversations. (Idit, age 42).*

## Theme 2: "From zero to a hundred": Challenges faced by counselors in returning to work after breast cancer recovery

The school counselors felt that the best medicine for them would be to return to work as soon as possible. Some of them described their place of work as their second home, a safe place where they feel wanted and needed. Prior to returning to work, most of them felt excitement mixed with fears of whether they would be able to function as they had in the past.

*Before I returned to work, I had many worries: Would I be able to maintain my staff and the students as I had done in the past? How would I be able to work a full day after being at home for so long? (Danielle, age 55).*

The counselors also described their difficulties in re-entering the workplace. They expected a warm and considerate attitude from their employers, but were met with a lack of flexibility. Their employers expected them to return immediately to the unending rat race and intensity of their jobs. If they asked for a break or expressed any weakness, they were met with verbal abuse and made to feel they were "exploiting" their status as cancer survivors. The counselors also felt disappointed and isolated when many of their counselor colleagues kept their distance and did not express any interest in what they had gone through.

*On the one hand, I was really pleased to return to work and felt I was doing something meaningful. On the other hand, I felt I needed to hold my head above water at work. My return was so intensive that I felt I was drowning. The administration did not cut me any slack (Amy, age 43).*

Participants reported experiencing numerous long-term symptoms stemming from their illness and treatment, particularly among school counselors who returned to work shortly after their treatment. These counselors described various issues, including fatigue, memory problems, and physical limitations. They stated that these symptoms had a negative impact on the quality of their work. They were frustrated because they felt they were disappointing themselves and their colleagues.

*I will never be what I once was. I tire easily, I have problems remembering and concentrating, and all of this affects how I function at work. My body is betraying me (Gabrielle, age 32).*

The school counselors who took part in the study expressed the difficulties they encountered during the transition from being at home to returning to work. Many of them described experiencing lingering symptoms, with fatigue being the most commonly reported. This fatigue significantly burdened the counselors and hindered their ability to perform their work effectively.

*When I initially resumed work, I felt more like a decoration than a counselor during that period. It was more for my own sake, to create a sense of safety in my workplace. However, I couldn't function properly due to fatigue. It took me some time to adjust. (Moran, age 31)*

The counselors also mentioned other challenges they faced, including memory difficulties and physical limitations. They shared that these symptoms had a negative impact on their work ability and quality. As a result, they had to develop strategies personally to overcome these difficulties. At times, the challenges became even more overwhelming, causing significant frustration among the counselors and affecting their relationships with colleagues.

*I will never be the person I was before. I experience problems with memory, concentration, extreme fatigue, and even hearing loss. All these factors affect my performance at work. (Noa, age 32)*

### Theme 3: "It's hard to listen to counselees' problems when I am immersed in my own crisis": How surviving breast cancer affects return to work among school counselors

Upon returning to work, participants reported they were less patient than in the past and had more difficulty containing the students they were trying to help.

*I came with eager to assist and support, but also feeling empty from my own problems. It was challenging for me to introduce additional difficulties there. (Tali, age 53)*

The counsellors described how the students' suffering and pain seemed disproportionate, and some of the problems appeared marginal and insignificant to them.

*I remember an incident when an 11th-grade student approached me, crying bitterly because the teacher had deducted 2 points from her test, which she felt was unnecessary. At that moment, I couldn't help but think, really*?! *I tried my best not to be judgmental, but it was almost impossible not to consider what I had been through in these past months, and now two points* (Hadas, 55)

The school counsellors expressed feelings of guilt for not being as inclusive, free, understanding, and patient as they were in the past. They were afraid to admit this to their superiors as they felt it would harm their professional abilities. One counselor shared a particularly challenging encounter with a student who sought her help but felt she hadn't provided sufficient support. She felt incapable of maintaining professionalism, and it tormented her deeply:

I remember one day when a student came to my office, a very problematic student with behavioral issues. He cursed me and then said, "I hope you die of cancer." I simply could not handle this and I responded in a way that was not appropriate for a counselor. He touched such a sensitive spot that I was in shock. I remember being very angry with him and speaking to him inappropriately (Anna, age 44).

## Discussion

This study sought to enhance our understanding of what school counselors experience upon returning to work after recovering from breast cancer. The participating counselors expressed their sadness upon being forced to stop working in order to focus on their recovery. They described their longing for school during treatment, where they felt needed and important. In one study conducted among cancer patients, including breast cancer patients, participants reported social isolation, frustration, despair and emotional distress [5–7, 9, 10]. Other studies indicate that some cancer patients continued going to their place of work in order to preserve their sense of professional identity [18–20]. The development of a professional identity plays a major role in defining the identity of each individual and serves as a commitment to the workplace [31]. When the counselors in the current study were forced to stop going to work, they conveyed that they felt their professional identity was being undermined and that they no longer belonged to the social circle that until their cancer diagnosis had been a familiar place.

The current study also found that breast cancer survivors were concerned about losing their jobs and felt betrayed when a replacement was hired. Research shows that cancer survivors looked forward to the first opportunity to return to work despite their problems in functioning [10, 32]. Survivors work routine is threatened after their diagnosis, and most express a sense of being out of control, uncertainty about their future and concerns about their ability to work and about the possibility of losing their jobs during the treatment period and after their recovery [32, 33]. Strategies to involve survivors during their treatment and/or recovery may help to improve communication during this difficult period, and reduce negative feelings as expressed by survivors [34].

The participants in the current study reported that upon returning to work they found that their functional capacity had decreased significantly, forcing them to make adjustments in

their jobs. Some participants preferred part-time jobs, longer-than-usual breaks, or offices on lower floors to avoid navigating stairs in high-rise buildings. A study conducted in Israel among 410 breast cancer survivors found that 33% of those who had worked prior to their diagnosis did not return to work, and among those who did, 48% limited their working hours, and only 19% did not make any changes in their work conditions [35]. This study also found that the chances for breast cancer survivors to enter the ranks of the unemployed are higher than for healthy women who never had cancer [36].

The participants in the current study reported that their employers scorned them and expressed disappointment in them, without showing any empathy or flexibility. Survivors reported that they felt they were a burden on the system and that too much was being expected of them. The level of support provided by the employer can be crucial in returning to the workplace in that it can either promote or hinder an employee's return and ability to function [37, 38]. Other studies reinforce this finding, showing that impatience on the part of a manager can have a negative impact on cancer survivors' performance and meaning of work, when they return to work [6, 18, 39, 40]. A study by de Rijk et al. [17] found that at workplaces where employers showed empathy, cancer survivors reported a sense of well-being and a desire to return to work as soon as possible. In contrast, when employers did not show flexibility, survivors tended to change jobs more frequently because they lacked a sense of belonging and well-being at work. Early discussions between survivors and their employers may help to establish more realistic expectations upon their return to work, and to reduce their negative experiences.

In the current study, the women described long-term symptoms from their illness and their treatment, such as fatigue, memory problems, physical limitations and more. Fatigue was generally perceived to be the most common physical barrier to return to work [41]. It is widely recognized that individuals diagnosed with cancer encounter diverse long-term symptoms (such as fatigue, pain, and neuropathy) that exhibit variations based on the specific type of cancer, particularly as a result of treatment-related side effects [21, 32, 42]. An individuals' physical limitations can impact their work performance, necessitating modifications to their roles, reduced work hours, adjustments in work methods, or even cessation of work [39, 40]. The presence of uncertainty surrounding the trajectory of the illness adds to the challenge of implementing these modifications effectively [43] and warrants further investigation.

The participants of the current study expressed a desire to return to work. Research indicates that the motivations for returning to work can be both internal and external [21, 44]. On a personal level, many women report that work provides them with a sense of purpose and reconnects them with their personal, professional, and social identities. Regarding external motivations, financial independence and the preservation of job security are predominantly cited [39]. The women in the current study stated that they experienced a reduced sense of emotional availability and patience towards their students in their capacity as school counselor. Therapists, when confronted with their own stress and challenges amid a crisis, are additionally exposed to the traumatic experiences shared by their clients [45, 46]. A consequential outcome for therapists in such circumstances is the phenomenon of vicarious traumatization, which denotes the cumulative and detrimental effects experienced by therapists who empathetically engage with clients who have been through traumatic events [47]. Prior research has established a link between vicarious traumatization and adverse mental health outcomes, compromised quality of therapeutic relationships, and shifts in worldview [48, 49]. As a result, therapists' encounters with vicarious traumatization may also impede the effectiveness of their treatment interventions [50].

To our knowledge, no studies have been conducted specifically examining the return to work experiences of school counselors recovering from breast cancer. The International

Classification of Functioning, Disability and Health (ICF) [51] is a comprehensive framework that addresses the various dimensions of health for individuals, although its primary emphasis has been on the impairment aspect of disabilities. The ICF model recognizes work as a pivotal societal role and highlights the influence of health conditions, environmental factors, and personal factors on an individual's participation return to work after cancer [40]. The observed difficulty faced by counselors in providing assistance to students as effectively as before, as reported in the present study, could potentially be explained by the proposed model. It is plausible that a combination of factors, including emotional and psychological states, organizational managerial responses, and personal characteristics, influenced their experiences upon returning to work.

## Practical recommendations

In the field of education, we recommend that educators, including school principals, receive training to expand their awareness regarding colleagues who have returned to work after recovering from cancer, an illness that harms both survivors' physical and emotional functioning. The training should ideally provide information about cancer survivors, symptoms, and suggest recommended means of acting flexibly and empathically in order to ease survivors' return to work, while discussing what survivors can expect and need from the organization. School counselors who are breast cancer survivors need to feel their colleagues' concern and feel they belong to the organization. Ongoing support is likely to promote survivors' emotional well-being and help them return to their professional positions as well as improve their performance at work.

## Study limitations

This study offers an in-depth understanding of the experiences of school counselors who are breast cancer survivors. The small size of the sample limits the generalizability of the findings to the broader breast cancer survivor population. Participant recruitment was carried out via convenience and snowball sampling methods on social networks, which may introduce bias. In addition, this online design introduced bias in the sample towards population groups with digita literacy or access to digital resources and those who are more socially connected, at least in the virtual sense. Furthermore, the recruitment method through a Facebook support group may not fully represent all breast cancer survivors, which also poses a limitation.

## Conclusions

School Counselors who are breast cancer survivors highlighted several challenges they face upon returning to work particularly the disruptions in work continuity, the impact on their professional identity and self-esteem, difficulty meeting employers' expectations and lack of flexibility and understanding from colleagues and managing physical symptoms and making adjustments. Study findings highlight the unique needs of survivors as they return to work both during treatment and after their return. Increased communication and the development of training for school staff may help to raise awareness and ultimately support survivors and improve their return to work experience.

## Acknowledgments

We thank all who participated in their efforts.

## Author Contributions

**Conceptualization:** Inbar Levkovich.

**Formal analysis:** Inbar Levkovich.

**Methodology:** Inbar Levkovich.

**Resources:** Lahav Rosman.

**Supervision:** Inbar Levkovich.

**Writing – original draft:** Inbar Levkovich.

**Writing – review & editing:** Christina Signorelli.

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
