## [Decision Letter · Decision Letter 0]

7 Nov 2023

PONE-D-23-15124"Hearing the Pupils Voices Through My Own Struggles”: A qualitative study of return to work among School Counselors who are breast cancer survivorsPLOS ONE

Dear Dr. Levkovich,

Thank you for submitting your manuscript to PLOS ONE. After careful consideration, we feel that it has merit but does not fully meet PLOS ONE’s publication criteria as it currently stands. Therefore, we invite you to submit a revised version of the manuscript that addresses the points raised during the review process.

We look forward to receiving your revised manuscript.

Kind regards,

Michal Ptaszynski, PhD

Academic Editor

PLOS ONE

Journal Requirements:

Reviewers' comments:

Reviewer's Responses to Questions

**Comments to the Author**

1. Is the manuscript technically sound, and do the data support the conclusions?

Reviewer #1: Partly

Reviewer #2: Yes

Reviewer #3: Yes

2. Has the statistical analysis been performed appropriately and rigorously? 

Reviewer #1: N/A

Reviewer #2: N/A

Reviewer #3: N/A

3. Have the authors made all data underlying the findings in their manuscript fully available?

Reviewer #1: No

Reviewer #2: Yes

Reviewer #3: Yes

4. Is the manuscript presented in an intelligible fashion and written in standard English?

Reviewer #1: Yes

Reviewer #2: Yes

Reviewer #3: Yes

5. Review Comments to the Author

Reviewer #1: This manuscript by Levkovich et al. examined the return-to-work experience among school counselors who are breast cancer survivors. The manuscript presents interesting perspectives from this occupation group and the qualitative methodology is well-described. However, the justification for the focus on this occupation group and discussion of results could be improved. My detailed comments are as follows.

Major comments

1. The research gap was highlighted as “To the best of our knowledge, no research has examined school counselors in this context.” (page 4). The lack of focus on a specific occupation is not a compelling rationale for this study. Instead, the authors should consider highlighting the characteristics of school counselors that make returning to work a more challenging task. There were some descriptors in the same paragraph, but they could be better rearranged. For instance, are physical and psychosocial functioning levels exceptionally crucial for this occupation? If so, late and long-term side effects may have an amplified impact on this group of survivors.

2. While the authors highlighted the ‘generalizability’ of study findings as a limitation, I would recommend a more in-depth reflection and discussion of this point. First, the term ‘generalizability’ is more associated with quantitative research. Instead, the synonymous term for qualitative research would be the ‘transferability’ of study findings. Second, the extent of transferability of findings requires a rich description of the context for the current study (i.e., school counselors in Israel). Hence, the authors should elaborate on the context of the target group, especially in the specific occupation role. Some examples of questions that may affect transferability: 1) What are the typical workload/ working hours of a school counselor? 2) What type of schools (e.g., private/ public) do school counselors work in? 3) Who are their target audience? Students of what age/ grade? How about parents? 4) Any occupation protection laws or policies that are protective/ discriminatory against patients undergoing cancer treatment and recovery? Additionally, the authors should also reflect on how transferable or similar the experiences will be with other occupations in related fields (e.g., education) or those requiring similar high demand for physical and psychosocial functioning.

3. Besides the participant characteristics described on page 5, there are some other relevant data that could enrich readers’ understanding of the studied population. Wherever available, the authors should consider reporting the following data: 1) timing of return to work relative to the time of remission, 2) types of schools worked at, 3) elapsed time since return to work. These additional data are crucial to reflect the phase of return to work – initial adjustment or extended period after settling in. Otherwise, it is unclear whether the findings (especially theme 3) are reflective of the difficulties of adjusting back to work or whether the psychosocial impact on daily working has persisted over several years. Consequently, this distinction will affect results interpretation and implications downstream.

4. The recruitment strategy was through social media outlets (page 5). Did the authors consider additional strategies to increase outreach to school counselors like snowballing? Are there existing professional societies or relevant training organizations that may have a rich network of contacts? Participants who responded through social media typically have extreme experiences (good/ bad) and have a desire to share. Consequently, school counselors with ‘moderate’ or more ‘neutral’ experiences may not have been well-sampled to capture their perspectives. The authors should reflect on the adequacy of the recruitment strategy and discuss it as a potential limitation of this study.

5. While the results reported a range of difficulties encountered by school counselors returning to work, there is a disproportionate lack of reporting on the coping strategies employed by the participants. With most challenges reported being consistent with the literature, the novelty and additional value of this study could highlight how well (or poorly) survivors are coping with the challenges, an inherent part of their return-to-work experience. For instance, were there questions related to understanding how the participants prepare themselves for return to work? Did they keep in contact with school colleagues over the treatment period? Did they negotiate for work rearrangements (e.g., part-time)? These are valuable data for analysis and reporting if available.

6. The practical recommendations (page 17) are all not survivor-centric and overemphasize the need for change on external factors vs. survivors’ self-regulation. While there is merit in recommending improvements in the work environment, how feasible or viable would training be in this context and under the Israeli education system? Furthermore, there is a lack of discussion on strategies or recommendations to improve/ maintain communication between survivors and their colleagues/ supervisors from diagnosis through treatment. Lastly, the current recommendations do not address theme 3. It appears there should be strategies or measures to support the psychological aspects of return to work, especially in this occupation group. For example, available counseling or psychosocial services may better support this group as they explore and reflect on their psychological/ emotional challenges when transitioning back to work.

Minor comments

1. Who are “school consultants” being referred to on page 11? Is this term synonymous with school counselors?

2. It was mentioned that “The participants in the current study reported that upon returning to work they found that their functional capacity had decreased significantly, forcing them to make adjustments in their jobs.” (page 15). What specific adjustments did participants report making? The adjustments were unclear from the results.

3. Why are interviews conducted over the Zoom platform a study limitation (page 18)?

Reviewer #2: This study is interesting and remains a rare topic.

However, a few clarification in the method section are needed:

Did the interview only once among each participant? Since the design used a phenomenology, how can the interviewers ensure the exploration of the meaning? How did the authors apply trustworthiness?

Reviewer #3: This study deals with the return to work of those who survive breast cancer and carry out a particularly delicate task, in contact with students and with many stakeholders. The authors correctly point out that return to work (RTW) of BCS is of great importance for quality of life and is associated with increased survival but is accompanied by a series of health problems that interfere with work capacity.

1. One aspect that can be critical in qualitative research on BCS RTW is time. Experiences related to return may be reported differently by those who have returned a long time ago and those who are returning now. The authors do not tell us whether they considered this issue.

2. The authors reported that the interviews observed the requirement imposed by the COVID-19 pandemic. An aspect that must be considered when evaluating the results is chronological: did the RTW occur during the pandemic, or before it? The difference is important, because in many countries the pandemic has imposed limitations on the possibility of working in direct contact with users for fragile workers, among whom people with breast cancer are generally included. An Italian qualitative study considered the case of a BCS teacher who was prevented from returning to work with students because she was fragile. It would be interesting to compare this situation with what was observed in Israel.

3. The first two of the themes that emerge from the thematic analysis are present in recent works on the same topic; for example, employer representatives feel the need to offer increased empathy and flexibility to BCS, offering personalized solutions [Bilodeau K, Gouin MM, Fadhlaoui A, Porro B. Supporting the return to work of breast cancer survivors: perspectives from Canadian employer representatives. J Cancer Surviv. 2023 May 4:1–9. doi: 10.1007/s11764-023-01382-5.] and this need is felt by the workers [Viseux M, Johnson S, Roquelaure Y, Bourdon M. Breast Cancer Survivors' Experiences of Managers' Actions During the Return to Work Process: A Scoping Review of Qualitative Studies. J Occup Rehabil. 2023 Mar 31. doi: 10.1007/s10926-023-10101-x.]. In a recent study, the BCS observe that not only can the rigid attitude of companies hinder RTW due to the difficulty in immediately providing the quantity of work previously provided, but also that a compassionate attitude can end up marginalizing the woman and excluding her from the professional evolution [Magnavita N. et al. Supporting Return to Work after Breast Cancer: A Mixed Method Study. Healthcare 2023, 11, 2343. https://doi.org/10.3390/healthcare11162343]. The authors could discuss these studies and compare them with their findings.

4. The authors should pay more attention to the third theme, relating to the willingness of BCS to devote themselves to others while being concerned about themselves. The motivational aspect is very important in reintegration into work. This aspect is what best characterizes their case history. they may find similar conditions in studies that have collected BCS from various work sectors or from healthcare.

5. In discussing the symptoms that BCS experience upon returning to work, such as fatigue, the authors could refer to the above reported mixed method study in which these problems, in addition to being derived from qualitative analysis, were measured in comparison with healthy women of similar age, demonstrating significant differences [Magnavita N. et al. Supporting Return to Work after Breast Cancer: A Mixed Method Study. Healthcare 2023, 11, 2343. https://doi.org/10.3390/healthcare11162343]. The theme of is often associated with other psychological problems [King et al.. Psychosocial experiences of breast cancer survivors: a meta-review. J Cancer Surviv. 2023 Mar 1. doi: 10.1007/s11764-023-01336-x.]

6. PLOS authors have the option to publish the peer review history of their article (what does this mean?). If published, this will include your full peer review and any attached files.

Reviewer #1: No

Reviewer #2: No

Reviewer #3: **Yes: **Nicola Magnavita

---

## [Author Response · Author response to Decision Letter 0]

29 Nov 2023

November 10, 2023

Prof. Michal Ptaszynski

Editor

PLOS ONE

Dear Professor Ptaszynski:

Submission of a Revision for Manuscript PONE-D-23-15124

Enclosed, please find our revised manuscript PONE-D-23-15124, titled “"Hearing the Pupils Voices Through My Own Struggles”: A qualitative study of return to work among School Counselors who are breast cancer survivors” which we are submitting for possible publication in PLOS ONE.

We thank you and the reviewers for the useful comments, which have significantly improved our manuscript. We have attached a version of the paper showing these changes, as well as a final “clean” copy of the manuscript. Below, we delineate the specific changes we made to address the reviewers’ concerns.

Sincerely, 

Prof. Inbar Levkovich, Ph.D.

Faculty of Graduate Studies

Oranim Academic College of Education, Israel

Dr. Christina Signorelli

School of Clinical Medicine, UNSW Medicine & Health, Discipline of Paediatrics, UNSW Sydney, Kensington, NSW, Australia

Kids Cancer Centre, Sydney Children’s Hospital, NSW, Randwick, Australia

REVIEWER #1:

This manuscript by Levkovich et al. examined the return-to-work experience among school counselors who are breast cancer survivors. The manuscript presents interesting perspectives from this occupation group and the qualitative methodology is well-described. However, the justification for the focus on this occupation group and discussion of results could be improved. My detailed comments are as follows.

Major comments

1. The research gap was highlighted as “To the best of our knowledge, no research has examined school counselors in this context.” (page 4). The lack of focus on a specific occupation is not a compelling rationale for this study. Instead, the authors should consider highlighting the characteristics of school counselors that make returning to work a more challenging task. There were some descriptors in the same paragraph, but they could be better rearranged. For instance, are physical and psychosocial functioning levels exceptionally crucial for this occupation? If so, late and long-term side effects may have an amplified impact on this group of survivors. 

Response: We agree that our rationale would benefit from a more explicit description of the unique psychosocial demands of the counselling profession. As suggested, we have re-arranged this paragraph and hope this has communicated our rationale more clearly. 

Please refer to the Introduction, page 5: 

“The current study examined the perspectives of school counselors who survived breast cancer and returned to the workplace. The job of school counselor is multidimensional and includes working individually with students and parents as well as enlisting the help of the entire school staff, and relies on psychosocial functioning to perform in this role [24]. Because of the demands of their job, school counselors are under a great deal of stress at work. Tension and anxiety at work are liable to cause fatigue, problems in functioning and decreased emotional availability [25]. Despite their increased risk of work stress, to the best of our knowledge, no research has examined school counselors in this context.”

2. While the authors highlighted the ‘generalizability’ of study findings as a limitation, I would recommend a more in-depth reflection and discussion of this point. First, the term ‘generalizability’ is more associated with quantitative research. Instead, the synonymous term for qualitative research would be the ‘transferability’ of study findings. Second, the extent of transferability of findings requires a rich description of the context for the current study (i.e., school counselors in Israel). Hence, the authors should elaborate on the context of the target group, especially in the specific occupation role. Some examples of questions that may affect transferability: 1) What are the typical workload/ working hours of a school counselor? 2) What type of schools (e.g., private/ public) do school counselors work in? 3) Who are their target audience? Students of what age/ grade? How about parents? 4) Any occupation protection laws or policies that are protective/ discriminatory against patients undergoing cancer treatment and recovery? Additionally, the authors should also reflect on how transferable or similar the experiences will be with other occupations in related fields (e.g., education) or those requiring similar high demand for physical and psychosocial functioning. 

Response: Thank you for the feedback. We have included information about the role of school counseling in Israel in the introduction on page 4. Additionally, we have incorporated details about the types of schools in which the research participants teach into the methodology section on page 6.

Please refer to the Introduction (page 4):

“To apply for a license in educational counseling in Israel, candidates must possess a master's degree in educational counseling and a teaching certificate, as per the guidelines of the Ministry of Education's Counseling Division (2023). Furthermore, school counselors must adhere to the code of ethics established by the Educational Counseling Board (Heled et al., 2019). School counselors are integral to the educational system, working full-time with students, educational staff, and parents. Their responsibilities extend beyond school hours, encompassing emergency situations, delivering lectures, and attending meetings. Although there is no specific policy for counselors recovering from illnesses such as cancer, those with over two years of service typically enjoy employment tenure protection under the Ministry of Education.”

Please refer to the Methods (page 6):

“All research participants were employed in public schools overseen by the Ministry of Education. Among the 28 participants, 11 served in high schools, 7 in middle schools, and 10 in elementary schools. Prior to their illness, all participants were full-time employeesI.”

3. Besides the participant characteristics described on page 5, there are some other relevant data that could enrich readers’ understanding of the studied population. Wherever available, the authors should consider reporting the following data: 1) timing of return to work relative to the time of remission, 2) types of schools worked at, 3) elapsed time since return to work. These additional data are crucial to reflect the phase of return to work – initial adjustment or extended period after settling in. Otherwise, it is unclear whether the findings (especially theme 3) are reflective of the difficulties of adjusting back to work or whether the psychosocial impact on daily working has persisted over several years. Consequently, this distinction will affect results interpretation and implications downstream. 

Response: Thank you for bringing this to our attention. We have incorporated the requested information regarding the timing of return to work relative to remission and additional relevant data into the manuscript, as suggested in your comments under "Research Participants" on page 5.

4. The recruitment strategy was through social media outlets (page 5). Did the authors consider additional strategies to increase outreach to school counselors like snowballing? Are there existing professional societies or relevant training organizations that may have a rich network of contacts? Participants who responded through social media typically have extreme experiences (good/ bad) and have a desire to share. Consequently, school counselors with ‘moderate’ or more ‘neutral’ experiences may not have been well-sampled to capture their perspectives. The authors should reflect on the adequacy of the recruitment strategy and discuss it as a potential limitation of this study. 

Response: We appreciate this comment. Indeed, locating participants through social networks may have limitations. The researchers tried to contact the existing association in Israel, but due to the large number of studies conducted in it during the study period, we were unable to promote it via this method. Part of the recruitment was indeed done by snowball sampling, when interviewees referred us to their colleagues. We have added a reference to this topic both in the method and in the limitations of the study. 

5. While the results reported a range of difficulties encountered by school counselors returning to work, there is a disproportionate lack of reporting on the coping strategies employed by the participants. With most challenges reported being consistent with the literature, the novelty and additional value of this study could highlight how well (or poorly) survivors are coping with the challenges, an inherent part of their return-to-work experience. For instance, were there questions related to understanding how the participants prepare themselves for return to work? Did they keep in contact with school colleagues over the treatment period? Did they negotiate for work rearrangements (e.g., part-time)? These are valuable data for analysis and reporting if available. 

Response: Thank you for bringing this to our attention. In response to your important comment, we took the opportunity to go through the interviews again and added a different point of view in the first theme (page 11):

“A minority of the school counselors reported that while they managed to keep in touch, sustaining these connections over an extended period was challenging. Some counselors experienced feelings of support, concern, and warmth from others, whereas others perceived pity, fear, and a sense of distance, suggesting that the relationships felt obligatory rather than stemming from genuine concern. One counselor shared,

They would call occasionally to check on me, inquire if I needed anything, or ask how I was managing. It was comforting to know someone was thinking of me. At times, I sensed their pity through the phone, which made it difficult. Therefore, I sometimes opted to respond with WhatsApp messages when I didn't have the energy for phone conversations. (Idit, age 42).”

6. The practical recommendations (page 17) are all not survivor-centric and overemphasize the need for change on external factors vs. survivors’ self-regulation. While there is merit in recommending improvements in the work environment, how feasible or viable would training be in this context and under the Israeli education system? Furthermore, there is a lack of discussion on strategies or recommendations to improve/ maintain communication between survivors and their colleagues/ supervisors from diagnosis through treatment. Lastly, the current recommendations do not address theme 3. It appears there should be strategies or measures to support the psychological aspects of return to work, especially in this occupation group. For example, available counseling or psychosocial services may better support this group as they explore and reflect on their psychological/ emotional challenges when transitioning back to work. 

Response: We appreciate this feedback on the practical recommendations presented in this paper and have included your suggestion to address communication improvements in the workplace and make counselling services available to survivors. 

Please refer to the Discussion (page 19): 

“School counselors who are breast cancer survivors need to feel their colleagues’ concern and feel they belong to the organization. Additionally, training should aim to improve communication between school counselors and their colleagues throughout the course of cancer diagnosis, treatment and recovery. Ongoing support is likely to promote survivors’ emotional well-being and help them return to their professional positions as well as improve their performance at work. We also recommend that provisions are made for these school counsellors to access counseling services for themselves, to support their transition back to work.”

Minor comments

1. Who are “school consultants” being referred to on page 11? Is this term synonymous with school counselors? 

Response: The error has been rectified.

2. It was mentioned that “The participants in the current study reported that upon returning to work they found that their functional capacity had decreased significantly, forcing them to make adjustments in their jobs.” (page 15). What specific adjustments did participants report making? The adjustments were unclear from the results.

 Response: Thank you for your valuable feedback. We have elaborated on the consequences faced by school counselors upon their return to school on page 16-17:

“Some participants preferred part-time jobs, longer-than-usual breaks, or offices on lower floors to avoid navigating stairs in high-rise buildings.”

3. Why are interviews conducted over the Zoom platform a study limitation (page 18)? 

Response: Thank you for raising this valuable point. We agree that use of the Zoom platform has become a normalised part of modern society in 2023, with more people now accustomed to communicating via this platform. Despite there being some documented limitations to rapport building in the literature between interviewer and interviewee, this impact are likely to have been minimal and we have removed this a limitation.

REVIEWER #2:

This study is interesting and remains a rare topic.

However, a few clarification in the method section are needed:

Did the interview only once among each participant? Since the design used a phenomenology, how can the interviewers ensure the exploration of the meaning? How did the authors apply trustworthiness? 

Response: The interviews were one-on-one, which we have now clarified in the methods (page 7). In addition, we have elaborated on the meaningful exploration of the data in our analytical approach, including reflections on trustworthiness section on page 9.

Please refer to the Methods, (page 7): The qualitative data were collected through one-on-one, in-depth, semi-structured interviews, each conducted with an interviewee only once.

Page 7: Before conducting interviews, interviewers were required to engage in a reflective process. This involved introspecting their own identities, social standings, assumptions, and life experiences, including any related to their own illness or that of their family members. They also considered how these factors might influence their interactions with interviewees. This preparatory step was essential for maintaining integrity in the research process.

REVIEWER #3:

This study deals with the return to work of those who survive breast cancer and carry out a particularly delicate task, in contact with students and with many stakeholders. The authors correctly point out that return to work (RTW) of BCS is of great importance for quality of life and is associated with increased survival but is accompanied by a series of health problems that interfere with work capacity.

1. One aspect that can be critical in qualitative research on BCS RTW is time. Experiences related to return may be reported differently by those who have returned a long time ago and those who are returning now. The authors do not tell us whether they considered this issue. 

Response: Thank you for your feedback. We have included details regarding the timeline for returning to work in the methodology chapter. Additionally, in the findings chapter, we observed a difference between counselors towards the end of the treatments, and we have referenced this observation in the second theme.

2. The authors reported that the interviews observed the requirement imposed by the COVID-19 pandemic. An aspect that must be considered when evaluating the results is chronological: did the RTW occur during the pandemic, or before it? The difference is important, because in many countries the pandemic has imposed limitations on the possibility of working in direct contact with users for fragile workers, among whom people with breast cancer are generally included. An Italian qualitative study considered the case of a BCS teacher who was prevented from returning to work with students because she was fragile. It would be interesting to compare this situation with what was observed in Israel. 

Response: The Israeli education system resumed in-person learning in 2022. The interviews for this study w

---

## [Decision Letter · Decision Letter 1]

8 Jan 2024

PONE-D-23-15124R1"Hearing the Pupils Voices Through My Own Struggles”: A qualitative study of return to work among School Counselors who are breast cancer survivorsPLOS ONE

Dear Dr. Levkovich,

Thank you for submitting your manuscript to PLOS ONE. After careful consideration, we feel that it has merit but does not fully meet PLOS ONE’s publication criteria as it currently stands. Therefore, we invite you to submit a revised version of the manuscript that addresses the points raised during the review process.

We look forward to receiving your revised manuscript.

Kind regards,

Michal Ptaszynski, PhD

Academic Editor

PLOS ONE

Journal Requirements:

Reviewers' comments:

Reviewer's Responses to Questions

**Comments to the Author**

1. If the authors have adequately addressed your comments raised in a previous round of review and you feel that this manuscript is now acceptable for publication, you may indicate that here to bypass the “Comments to the Author” section, enter your conflict of interest statement in the “Confidential to Editor” section, and submit your "Accept" recommendation.

Reviewer #1: All comments have been addressed

Reviewer #3: All comments have been addressed

2. Is the manuscript technically sound, and do the data support the conclusions?

Reviewer #1: Yes

Reviewer #3: Yes

3. Has the statistical analysis been performed appropriately and rigorously? 

Reviewer #1: N/A

Reviewer #3: N/A

4. Have the authors made all data underlying the findings in their manuscript fully available?

Reviewer #1: No

Reviewer #3: Yes

5. Is the manuscript presented in an intelligible fashion and written in standard English?

Reviewer #1: Yes

Reviewer #3: Yes

6. Review Comments to the Author

Reviewer #1: The authors have addressed the comments well. Some minor points for clarification:

1) "The participants of the current study expressed a desire to return to work but also its consequences." (page 19): the current phrasing suggests that participants expressed a desire for return to work consequences. Is this accurate?

2) Under study limitations (page 21), suggest "results may not be transferrable to broader population" instead of 'generalized'. Also, it may be good to specify who constitutes the 'broader population'? Is it cancer survivor or breast cancer survivor population?

3) Under study limitations (page 21), the description of the sampling method can be refined to specify the sampling bias introduced. For instance, this way of sampling likely excludes counsellors who are 'off the radar', not as actively involved in professional groups or social media.

Reviewer #3: The manuscript has been revised, addressing all the changes suggested by the reviewers. There are no suggestions.

7. PLOS authors have the option to publish the peer review history of their article (what does this mean?). If published, this will include your full peer review and any attached files.

Reviewer #1: No

Reviewer #3: **Yes: **Nicola Magnavita

---

## [Author Response · Author response to Decision Letter 1]

10 Jan 2024

January 10, 2024

Prof. Michal Ptaszynski

Editor

PLOS ONE

Dear Professor Ptaszynski:

Submission of a Revision for Manuscript PONE-D-23-15124R1

Enclosed, please find our revised manuscript PONE-D-23-15124R1, titled “"Hearing the Pupils’ Voices Through My Own Struggles”: A Qualitative Study Of Return To Work Among School Counselors Who Are Breast Cancer Survivors” which we are submitting for possible publication in PLOS ONE.

We thank you and the reviewers for the useful comments, which have significantly improved our manuscript. We have attached a version of the paper that shows these changes, along with a final “clean” copy of the manuscript. Below, we delineate the specific modifications we made to address the reviewers’ concerns.

Sincerely, 

Prof. Inbar Levkovich, Ph.D.

Faculty of Graduate Studies

Oranim Academic College of Education, Israel

Dr. Christina Signorelli

School of Clinical Medicine, UNSW Medicine & Health, Discipline of Paediatrics, UNSW Sydney, Kensington, NSW, Australia

Kids Cancer Centre, Sydney Children’s Hospital, NSW, Randwick, Australia

Comments to the Author

1. If the authors have adequately addressed your comments raised in a previous round of review and you feel that this manuscript is now acceptable for publication, you may indicate that here to bypass the “Comments to the Author” section, enter your conflict of interest statement in the “Confidential to Editor” section, and submit your "Accept" recommendation.

Reviewer #1: All comments have been addressed

Reviewer #3: All comments have been addressed

2. Is the manuscript technically sound, and do the data support the conclusions?

Reviewer #1: Yes

Reviewer #3: Yes

3. Has the statistical analysis been performed appropriately and rigorously? 

Reviewer #1: N/A

Reviewer #3: N/A

4. Have the authors made all data underlying the findings in their manuscript fully available?

Reviewer #1: No

Thank you for the feedback. in accordance with the rules of the Ethics Committee and the Helsinki Committee and considering that our research involved personal interviews with women in cancer recovery, weregret that we are unable to share this information. Consequently, we have maintained the anonymity of the participants and have refrained from presenting their information in detail. We added our statement regarding data availability as follows: 

Availability of data and materials: Data cannot be shared beyond individual quotations in the manuscript due to participants’ lack of consent for the publication of their full transcripts. The interviews contain sensitive data, and this decision aligns with the ethics committee of Oranim College. Please contact Miss Shaulov, administrative coordinator of the research authority in Oranim College for any further information: research@oranim.ac.il

Reviewer #3: Yes

5. Is the manuscript presented in an intelligible fashion and written in standard English?

Reviewer #1: Yes

Reviewer #3: Yes

6. Review Comments to the Author

Reviewer #1: The authors have addressed the comments well. Some minor points for clarification:

1) "The participants of the current study expressed a desire to return to work but also its consequences." (page 19): the current phrasing suggests that participants expressed a desire for return to work consequences. Is this accurate?

Thanks for the feedback. We re-worded the sentence:

Page 19: “The participants of the current study expressed a desire to return to work.”

2) Under study limitations (page 21), suggest "results may not be transferrable to broader population" instead of 'generalized'. Also, it may be good to specify who constitutes the 'broader population'? Is it cancer survivor or breast cancer survivor population?

Thank you. Here is how we have revised the wording of the sentence:

Page 21: “The small size of the sample limits the generalizability of the findings to the broader breast cancer survivor population”.

3) Under study limitations (page 21), the description of the sampling method can be refined to specify the sampling bias introduced. For instance, this way of sampling likely excludes counsellors who are 'off the radar', not as actively involved in professional groups or social media.

Thank you for pointing this out.

Page 21: “In addition, this online design introduced bias in the sample towards population groups with digita literacy or access to digital resources and those who are more socially connected, at least in the virtual sense. Furthermore, the recruitment method through a Facebook support group may not fully represent all breast cancer survivors, which also poses a limitation."

Reviewer #3: The manuscript has been revised, addressing all the changes suggested by the reviewers. There are no suggestions.

7. PLOS authors have the option to publish the peer review history of their article (what does this mean?). If published, this will include your full peer review and any attached files.

Do you want your identity to be public for this peer review? For information about this choice, including consent withdrawal, please see our Privacy Policy.

Reviewer #1: No

Reviewer #3: Yes: Nicola Magnavita

We would like to express our gratitude once again to the reviewers for their generous assistance, which has significantly contributed to the improvement of our manuscript. We hope that the reviewers now find our manuscript suitable for publication in PLOS ONE.

---

## [Decision Letter · Decision Letter 2]

27 Feb 2024

"Hearing the Pupils Voices Through My Own Struggles”: A qualitative study of return to work among School Counselors who are breast cancer survivors

PONE-D-23-15124R2

Dear Dr. Levkovich,

We’re pleased to inform you that your manuscript has been judged scientifically suitable for publication and will be formally accepted for publication once it meets all outstanding technical requirements.

Kind regards,

Michal Ptaszynski, PhD

Academic Editor

PLOS ONE

Additional Editor Comments (optional):

Reviewers' comments:

Reviewer's Responses to Questions

**Comments to the Author**

1. If the authors have adequately addressed your comments raised in a previous round of review and you feel that this manuscript is now acceptable for publication, you may indicate that here to bypass the “Comments to the Author” section, enter your conflict of interest statement in the “Confidential to Editor” section, and submit your "Accept" recommendation.

Reviewer #1: All comments have been addressed

2. Is the manuscript technically sound, and do the data support the conclusions?

Reviewer #1: Yes

3. Has the statistical analysis been performed appropriately and rigorously? 

Reviewer #1: N/A

4. Have the authors made all data underlying the findings in their manuscript fully available?

Reviewer #1: Yes

5. Is the manuscript presented in an intelligible fashion and written in standard English?

Reviewer #1: Yes

6. Review Comments to the Author

Reviewer #1: The authors have addressed the comments well with additional revisions where appropriate. No further comments.

7. PLOS authors have the option to publish the peer review history of their article (what does this mean?). If published, this will include your full peer review and any attached files.

Reviewer #1: No
